# The Role of microRNAs Related to Apoptosis for *N*-Methyl-d-Aspartic Acid-Induced Neuronal Cell Death in the Murine Retina

**DOI:** 10.3390/ijms25021106

**Published:** 2024-01-16

**Authors:** Kohei Sone, Asami Mori, Kenji Sakamoto, Tsutomu Nakahara

**Affiliations:** Department of Molecular Pharmacology, Kitasato University School of Pharmaceutical Sciences, 9-1 Shirokane 5-Chome, Minato-ku, Tokyo 108-8641, Japan; sone@rohto.co.jp (K.S.); mori@pharm.teikyo-u.ac.jp (A.M.); nakaharat@pharm.kitasato-u.ac.jp (T.N.)

**Keywords:** retina, microRNA, excitotoxicity, retinal ganglion cell, apoptosis, MCL-1, Bax, Bim

## Abstract

Glaucoma is one of the leading causes of acquired blindness and characterized by retinal ganglion cell (RGC) death. MicroRNAs are small noncoding RNAs that degrade their target mRNAs. Apoptosis is one of the common mechanisms leading to neuronal death in many neurodegenerative diseases, including glaucoma. In the present study, we identified microRNAs that modulate RGC death caused by the intravitreal injection of *N*-methyl-d-aspartic acid (NMDA). We found an upregulation of miR-29b and downregulation of miR-124 in the retina of the NMDA-injected eyes. The intravitreal injection of an miR-29b inhibitor 18 h before NMDA injection reduced RGC death and the downregulation of myeloid cell leukemia 1 (MCL-1), an anti-apoptotic factor, induced by intravitreal NMDA. The intravitreal injection of an miR-124 mimic 18 h before NMDA injection also reduced RGC death and the upregulation of B-cell/chronic lymphocytic leukemia lymphoma 2 (bcl-2)-associated X protein (Bax) and bcl-2 interacting protein (Bim), pro-apoptotic factors, induced by intravitreal NMDA. These data suggest that expressional changes in microRNA are involved in the excitotoxicity of RGCs, and that complement and/or inhibition of microRNA may be a potential therapeutic approach for the diseases related to the excitotoxicity of RGCs, such as glaucoma and retinal central artery occlusion.

## 1. Introduction

A degeneration of retinal ganglion cells (RGCs) is characteristically observed in the glaucomatous retinas [1]. However, the mechanisms of RGC death in glaucoma are not completely understood. An influx of high amounts of Ca^2+^ following the activation of the *N*-methyl-d-aspartic acid (NMDA) receptor causes excitoneurotoxicity [2,3], which is thought to be involved in the mechanisms of RGC death observed in glaucoma and retinal central artery occlusion [4].

MicroRNAs (miRs) are single-stranded endogenous short noncoding RNAs whose length is approximately 21 to 25 nucleotides [5,6]. A mature miR interacts with the target mRNA and induces its degradation or translational inhibition. Therefore, miRs are important posttranscriptional regulators of gene expression and essential players in various cellular pathways. Recently, expressional changes in miR have been suggested to be related to the pathogenesis of neurodegenerative diseases [7,8]. The identification of miRs involved in the disease can lead to new approaches to therapy by either rescuing the downregulated miRs or inhibiting the upregulated miRs [9].

Recently, expressional changes in various miRs have been reported in the retina of animal models of glaucoma. For example, a downregulation of let-7a, miR-16, miR-25, miR-29b, miR-106b, miR-181c, miR-204, and miR-497 and an upregulation of miR-27a were reported in rats with high intraocular pressure [10]. However, there is little information about expressional changes in miRs in retinas subjected to excitotoxicity.

Previously, miR-29b and miR-124 were reported as miRs related to apoptosis. An upregulation of miR-29b, which is related to myeloid cell leukemia 1 (MCL-1), an anti-apoptotic protein, was reported in the apoptotic cells induced by tumor necrosis factor-related apoptosis-inducing ligand [11]. miR-124 is expressed specifically in the central nervous system. In the retina, miR-124 is reported to be expressed in the ganglion cell layer, inner nuclear layer, and inner segment [12,13,14]. The neuroprotective effect of an miR-124 mimic via the downregulation of B-cell/chronic lymphocytic leukemia lymphoma 2 (bcl-2)-associated X protein (Bax) and bcl-2 interacting protein (Bim), pro-apoptotic proteins, was reported in a newborn rat with thyroid hypofunction [15] and Parkinson’s disease model mice [16].

The work is aimed at showing the correlation between the excitoneurotoxicity of the retinal neurons and expressional changes in miR-29b and miR-124. Based on the results, we examined whether the miR-29b inhibitor and miR-124 mimic changed the expression levels of their target mRNAs, and whether the miR-29b inhibitor and miR-124 mimic were protective against *N*-methyl-d-aspartic acid (NMDA)-induced retinal injury in mice.

## 2. Results

### 2.1. The Changes in the Expression of miR-29b and miR-124 in the Retinas of the NMDA-Injected Eyes in C57BL/6J Mice

We demonstrated the changes in the expression of miR-29b and miR-124 in the retinas of the NMDA-injected eyes of C57BL/6J mice using real-time reverse transcription polymerase chain reaction (RT-PCR). The summary of the relative expression of miR-29b and miR-124 before NMDA injection and 4, 8, 12, and 24 h after NMDA injection from four (8 h) or five (other time points) independent experiments is shown in Figure 1. The expression level of snoRNA234 was used as an internal standard. The expression level of miR-29b was increased to 1.71 ± 0.18 and 1.84 ± 0.08, 8 and 12 h after NMDA injection, respectively (Figure 1a). The expression level of miR-124 was decreased to 0.484 ± 0.06, 8 h after NMDA injection (Figure 1b). However, the expression level of miR-124 tended to increase 12 h after NMDA injection (*p* = 0.23). The expression level of miR-29b and miR-124 was restored 12 h after NMDA injection.

### 2.2. The Effects of the miR-29b Inhibitor and miR-124 Mimic on Retinal Injury Induced by Intravitreal NMDA in Transgenic Mice Expressing Enhanced Cyan Fluorescent Protein (ECFP) Specificity in RGCs In Vivo

We demonstrated the effects of the miR-29b inhibitor and miR-124 mimic on RGC death induced by intravitreal NMDA. Representative photomicrographs of the retinal flatmounts of the transgenic mice expressing enhanced cyan fluorescent protein (ECFP) specificity in RGCs (B6.Cg-Tg(Thy1-CFP)23Jrs/J) 7 days after NMDA injection are shown in Figure 2. As the negative control (NC), an miR inhibitor NC and miR mimic NC were used in the present study. Both the miR mimic NC and miR inhibitor NC used in the present study are based on cel-miR-67, which is a *Caenorhabditis elegans* miR and confirmed to have minimal sequence identity with murine miRNAs. They are confirmed to have no identifiable effects on the tested miRNA function by the manufacturer. The treatment of neither the miR inhibitor NC nor miR mimic NC 24 h before NMDA injection affected the NMDA-induced RGC death (Figure 2). The intravitreal treatment of the miR-29b inhibitor and miR-124 mimic 24 h before NMDA injection prevented RGCs induced by NMDA (Figure 2). The mean cell numbers of the ECFP-positive cells from four (miR mimic NC), five (miR inhibitor NC), or six (miR-29b inhibitor and miR-124 mimic) independent experiments are summarized in Figure 3.

### 2.3. The Effect of the miR-29b Inhibitor on the Apoptosis of Retinal Neuronal Cells Induced by Intravitreal NMDA in C57BL/6J Mice In Vivo

We demonstrated the effects of the miR-29b inhibitor on the apoptosis of retinal neuronal cells induced by intravitreal NMDA using terminal deoxynucleotidyl transferase-mediated dUTP nick-end labeling (TUNEL) staining. Representative photomicrographs of the retinal sections obtained 12 h after NMDA injection are shown in Figure 4A. the Intravitreal treatment of the miR inhibitor NC 24 h before NMDA injection did not affect the increase in the number of TUNEL-positive cells induced by intravitreal NMDA (Figure 4A, panels c and d). Intravitreal treatment of the miR-29b inhibitor 18 h before NMDA injection reduced the number of TUNEL-positive cells induced by intravitreal NMDA (Figure 4A, panels g and h). The mean numbers of the TUNEL-positive cells from five independent experiments are summarized in Figure 4B,C.

### 2.4. The Effect of the miR-29b Inhibitor on the Change in the Expression of MCL-1 in the Retina Induced by Intravitreal NMDA in C57BL/6J Mice In Vivo

We demonstrated the effects of the miR-29b inhibitor on the change in the expression of MCL-1 in the retina induced by intravitreal NMDA. Representative photomicrographs of the retinal sections obtained 12 h after NMDA injection are shown in Figure 5A. The intravitreal treatment of the miR inhibitor NC 18 h before NMDA injection did not affect the decrease in the number of MCL-1-positive cells induced by intravitreal NMDA (Figure 5A, panels c and d). The intravitreal treatment of the miR-29b inhibitor 18 h before NMDA injection reduced the decrease in the number of MCL-1-positive cells induced by intravitreal NMDA (Figure 5A, panels g and h). The mean numbers of MCL-1-positive cells from five independent experiments are summarized in Figure 5B.

### 2.5. The Effect of the miR-124 Mimic on the Apoptosis of Retinal Neuronal Cells Induced by Intravitreal NMDA in C57BL/6J Mice In Vivo

We demonstrated the effects of the miR-124 mimic on the apoptosis of retinal neuronal cells induced by intravitreal NMDA using TUNEL staining. Representative photomicrographs of the retinal sections obtained 12 h after NMDA injection are shown in Figure 6A. Intravitreal treatment of the miR mimic NC 18 h before NMDA injection did not affect the increase in the number of TUNEL-positive cells induced by intravitreal NMDA (Figure 6A, panels c and d). Intravitreal treatment of the miR-124 mimic 18 h before NMDA injection reduced the number of TUNEL-positive cells induced by intravitreal NMDA (Figure 6A, panels g and h). The mean numbers of the TUNEL-positive cells from five independent experiments are summarized in Figure 6B,C.

### 2.6. The Effect of the miR-29b Inhibitor on the Change in the Expression of Bax and Bim in the Retina Induced by Intravitreal NMDA in C57BL/6J Mice In Vivo

We demonstrated the effects of the miR-124 mimic on the change in the expression of Bax and Bim in the retina induced by intravitreal NMDA. Representative photomicrographs of the retinal sections obtained 8 h after NMDA injection are shown in Figure 7 and Figure 8. Intravitreal treatment of the miR mimic NC 18 h before NMDA injection did not affect the increase in the expression of Bax (Figure 7A, panels d–f) and Bim (Figure 8A, panels d–f) in the ganglion cell layer (GCL) and the inner plexiform layer (IPL) in the retina induced by intravitreal NMDA. The positive signal of Bax and Bim in IPL was co-localized with that of GS, a marker of Müller glial cells. Intravitreal treatment of the miR-124 mimic 18 h before NMDA injection reduced the increase in the expression of Bax (Figure 7A, panels j–l) and Bim (Figure 8A, panels j–l) induced by intravitreal NMDA. The mean fluorescence intensity of Bax and Bim from five independent experiments is summarized in Figure 7B,C and Figure 8B,C.

## 3. Discussion

Previously, the relationships between miR and diabetic retinopathy [17,18], oxygen-induced retinopathy [19], Müller glial cell function [20,21], light adaptation [22], and so on have been reported. Although α-Melanocyte-stimulating hormone was reported to reduce the excitotoxicity of chicken retinal neurons via the downregulation of miR-194 [23], little information has been shown about the relationship between miR and excitotoxicity of the mammalian retinal neuron.

In the present study, we first demonstrated that an upregulation of miR-29b and downregulation of miR-124 in the retina of the NMDA-injected eyes in the mice in vivo. Intravitreal injection of the miR-29b inhibitor and miR-124 mimic reduced RGC death. They reduced the downregulation of MCL-1, an anti-apoptotic factor, and upregulation of Bax and Bim, pro-apoptotic factors, in the retina induced by intravitreal NMDA. These results suggest that expressional changes in microRNA are involved in the excitotoxicity of RGCs, and that the miR-29b inhibitor and miR-124 prevented NMDA-induced retinal injury via a reduction in the downregulation of pro-apoptotic proteins and upregulation of pro-apoptotic proteins.

Our preliminary experiments showed that the apoptosis of the retinal neurons induced by intravitreal NMDA started to occur 4~8 h after NMDA injection (Appendix A). It has been reported that the loss of RGCs and an increase in the phosphorylation of p38 MAP kinase, a pro-apoptotic kinase, in the retina were observed 6 h after intravitreal NMDA injection in rats [24,25]. Therefore, the time course of apoptosis induced by intravitreal NMDA observed in the present study is consistent with that observed by other researchers. In the present study, the increase in the expression of miR-29b and the decrease in the expression of miR-124 were observed 8–12 h after the intravitreal NMDA injection. Because the time course of these expressional changes is consistent with that of apoptosis, it is possible that the expressional changes in these miRs are involved in the mechanisms of apoptosis of the retinal neurons induced by intravitreal NMDA.

Unfortunately, we could not clarify the mechanisms and/or the triggers of the expressional changes in miRs by intravitreal NMDA in the present study. Recently, the high-mobility group box-1 (HMGB1)–receptor of advanced glycation end-products (RAGE) axis has been found to be involved in the regulation of miR expression [26,27]. We previously reported that the HMGB1-RAGE axis was involved in neurodegeneration induced by intravitreal NMDA in the retina [28]. Although further studies are clearly needed, we consider the HMGB1-RAGE axis to be a possible candidate of the triggers of the expressional changes in miRs.

Although an upregulation of miR-29b was observed in the retina of the NMDA-injected eyes in the present study, Jayaram et al. reported a downregulation of miR-29b in the retina of the glaucoma model rats with high ocular pressure previously [10]. In the present study, a quantitative measurement of miR was carried out when neuronal apoptosis started to occur. In contrast, Jayaram et al. [10] quantified miRs five weeks after subjecting to high intraocular pressure, when the retinal neuronal cell loss had progressed. In addition, we observed that miR-181a, whose expression was reported to be decreased in the retina of the ischemia–reperfusion model induced by very high intraocular pressure [29], was not affected by intravitreal NMDA [30]. Therefore, the discrepancy of the expressional change in miRs may be caused by the time course of the quantification and/or the difference in the intraocular pressure.

In the present study, we used AteloGene Local Use “Quick Gelation” as the vehicle of the miRNA mimic/inhibitor and their NC. The manufacturer states that the reagent gelates by warming to approximately 37 °C and can be used for the sustained release of miRs. We confirmed that the gelated reagent remained in the intravitreal space when the retina was excised from the eyeball, suggesting that a sustained release of the miR mimic/inhibitor can be expected. Unfortunately, the duration of the release of the miR mimic/inhibitor is unknown. However, it is enough to observe the minimum neuroprotective effects of the miR-29b inhibitor and miR-124 mimic. Because neither the miR inhibitor NC nor the miR mimic NC was neuroprotective, intravitreal injection of nucleic acid and AteloGene Local Use “Quick Gelation” is not related to the neuroprotective effects found in the present study. An improvement in the reagent may enable a longer duration of the release of the miR mimic/inhibitor and more powerful neuroprotection. Further improvements and experiments are clearly needed to realize more notable neuroprotection.

The interaction between miR-29b and MCL-1 mRNA has been predicted by microRNA.org [31] and a previous report [11]. MCL-1 is an anti-apoptotic protein belonging to the Bcl-2 family [32] and regulates neuronal apoptosis [33]. MCL-1 reduces the activation of Bax and/or Bcl-2 homologous antagonist/killer (Bak), which are pro-apoptotic proteins and lead to cytochrome c release from mitochondria and then apoptosis [34]. It is possible that the neuroprotective effects of the miR-29b inhibitor found in the present study can be caused by a reduction in the degradation of MCL-1 mRNA induced by intravitreal NMDA, upregulation of the expression of MCL-1 protein, and then a reduction in apoptosis.

Similarly, the interactions between miR-124 and mRNA of Bax and Bim, pro-apoptotic proteins, have been predicted by microRNA.org [31] and a previous report [16]. It has been reported that miR-124 expresses in the ganglion cell layer, the inner nuclear layer, and the photoreceptor inner segment in the retina, and that miR-124 is involved in existence of the retinal neurons [14]. It has been reported that the upregulation of Bax starts to be seen 6 h after intravitreal NMDA in the retina [25,35], and that the upregulation of Bim starts to be seen 2 h after treatment with NMDA in the cerebellar granular neurons [35]. In addition, the knockout of Bax [25,35] and Bim [36] has been reported to be neuroprotective in the retina and the brain. Combining the results in the present study and the previous reports, the neuroprotective effects of the miR-124 mimic found in the present study could be caused by a reduction in apoptosis through preventing the upregulation of Bax and Bim by intravitreal NMDA. Because we found that an intravitreal injection of the miR-124 inhibitor alone slightly reduced the number of RGCs (Appendix A), it is suggested that miR-124 also contributes to the existence of the retinal neurons in the normal condition.

In conclusion, this is the first study to demonstrate that the miR-29b inhibitor and miR-124 mimic have neuroprotective effects against retinal injury induced by intravitreal NMDA via a reduction in apoptosis (Figure 9). Prevention of the downregulation of MCL-1 by the miR-29b inhibitor and that of the upregulation of Bax and Bim by the miR-124 mimic are possibly involved in the underlying mechanisms. The present study suggests that expressional changes in miR are involved in the excitotoxicity of RGCs, and that the rescue and/or inhibition of microRNA may be a potential therapeutic approach for diseases related to the excitotoxicity of RGCs, such as glaucoma and retinal central artery occlusion.

## 4. Materials and Methods

### 4.1. Animals

The experimental procedures in the current study adhered to the Regulations for the Care and Use of Laboratory Animals and were approved by the Institutional Animal Care and Use Committee of Kitasato University (protocol code: A09-4, E09-2 and T10-1, date of approval: 1 May 2017). Male and female C57BL/6J mice (7-8 weeks old, Charles River Japan, Yokohama, Japan) and B6.Cg-Tg (Thy1-CFP) 23Jrs/J mice (7–8 weeks old, The Jackson Laboratory, Bar Harbor, ME) were purchased and maintained by brother–sister mating in the animal room of Kitasato university. We reared the mice in the animal room kept at 25 °C with a 12 h:12 h light–dark cycle. The mice were fed and watered ad libitum.

### 4.2. Intravitreal Injection

For the anesthetization of the mice, intraperitoneal injection of the mixed solution of ketamine (90 mg/kg, i.p., Daiichi-Sankyo, Tokyo, Japan) and xylazine (10 mg/kg, i.p., Tokyo Kasei, Tokyo, Japan) was carried out. Intravitreal injection was carried out as previously described [28,37,38,39]. The NMDA-induced retinal damage was not affected by ketamine, an NMDA receptor antagonist, at the dose used in the present study [37]. A 33-gauge needle connected to a 25 µL microsyringe (MS-N25, Ito Seisakujo, Fuji, Japan) was used for intravitreal injection. One microliter of the drug solution was injected into one eye. The solution without NMDA was injected into the contralateral eye, which served as the control.

### 4.3. Real-Time RT-PCR

The mice were sacrificed 4, 8, 12, and 24 h after NMDA injection. The eyes were enucleated, and the retinas were isolated. TRIZOL reagent (Invitrogen, Carlsbab, CA, USA) was used for total RNA isolation following the manufacturer’s instructions. The Mir-X miRNA First-Strand Synthesis Kit (Clontech, Mountain View, CA, USA) was used for first-strand cDNA template synthesis. The KAPA SYBR Fast qPCR Kit (KAPA Biosystems, Wilmington, MA, USA), target-specific 5′-primers, and mRQ 3′ primer from the Mir-X miRNA First-Strand Synthesis Kit were used for SYBR Green real-time RT-PCR reaction. The sequences of the target-specific 5′-primers are summarized in Table 1. Mouse snoRNA234 was used as an internal control. The delta–delta Ct method was used for calculating relative gene expression values [40].

### 4.4. Drug Preparations and Treatment

Intravitreal injection was carried out as previously described [38,39]. NMDA (Nacalai Tesque, Kyoto, Japan) was dissolved in saline at 4.0 × 10^−2^ M. miRIDIAN microRNA Mimic Negative Control #1 (CN-001000-01-05, Dharmacon, Lafayette, CO, USA), miRIDIAN microRNA Hairpin Inhibitor Negative Control #1 (IN-001005-01-05, Dharmacon), miRIDIAN microRNA Mouse mmu-miR-29b-1-5p-Hairpin Inhibitor (IN-311005-02-0002,Dharmacon), and miRIDIAN microRNA Mouse mmu-miR-124-3p-Mimic (C-310391-05-0002, Dharmacon) were solved with AteloGene Local Use “Quick Gelation” (KOKEN, Tokyo, Japan) at 10 pmol/µL. They were treated intravitreally 18 h before intravitreal injection of NMDA.

### 4.5. Retinal Flatmount Preparation

The methods to make a retinal flatmount have been described previously [38,39]. The mice were sacrificed 7 days after NMDA injection. The eyes were enucleated and fixed with 4% paraformaldehyde in 0.1M phosphate-buffered saline (Nacalai Tesque) for 1 h at 4 °C. A confocal laser microscope (LSM710, Carl Zeiss, Oberkochen, Germany) was used to collect the images.

The number of neurons expressing ECFP was manually counted using image-processing software (Adobe Photoshop CS6, San Jose, CA, USA) in the area of 0.2 mm^2^ that was 500 μm away from the optic disc. The cell survival rate was calculated as follows.
$$Cell\;survival\;rate=\frac{The\;number\;of\;ECFP-positive\;cells\;in\;the\;retina\;of\;the\;NMDA-injected\;eye}{The\;number\;of\;ECFP-positive\;cells\;in\;the\;retina\;of\;the\;contralateral\;eye}$$

### 4.6. Preparation of the Retional Sections

Histological evaluation methods have been described previously [28,38,39]. Briefly, animals were sacrificed with an overdose of pentobarbital sodium 12 h after the intravitreal injection of NMDA. The enucleated eyes were fixed with a Davidson solution (37.5% ethanol (Nacalai Tesque), 9.3% paraformaldehyde (Nacalai Tesque), and 12.5% acetic acid (Nacalai Tesque)) for 24 h at room temperature. Fixed eyes were embedded in paraffin, and 4 µm thickness horizontal sections through the optic nerve head were cut with a microtome (HM325, Microm International, Walldorf, Germany) and a microtome blade (PATH BLADE + PRO by Kai, Matsunami Glass, Kishiwada, Japan).

### 4.7. TUNEL Staining and Immunohistochemistry

Retinal sections were made as described above.

TUNEL staining was carried out to detect apoptotic cells using the ApopTag^®^ Plus Fluorescein In Situ Apoptosis Detection Kit (Chemicon, Temecula, CA, USA) according to the manufacturer’s instructions.

The method for immunohistochemical analysis was described previously [28]. A solution of 10% normal goat serum in PBS (anti-MCL-1 antibody) or Blocking One Histo (anti-Bax antibody and anti-Bim antibody; Nacalai Tesque) was used for blocking. Then, retinal sections were incubated in rabbit polyclonal anti-MCL-1 antibody (1:100; ab32087, Abcam, Cambridge, UK), rabbit polyclonal anti-Bax antibody (1:250; ab32503, Abcam), and rabbit polyclonal anti-Bim antibody (1:100; ab32158, Abcam) overnight at 4 °C. Finally, sections were treated with Alexa Fluor^®^ Plus 488 goat anti-rabbit IgG (1:200, Invitrogen) for 1 h at room temperature. For co-immunostaining to detect glutamine synthetase, a marker of Müller glial cells, mouse monoclonal anti-glutamine synthetase antibody (1:1000, MAB302. Merck, Darmstadt, Germany) and Alexa Fluor^®^ Plus 647 goat anti-mouse IgG (1:200, Invitrogen) were used. Can Get Signal Immunostain Immunoreaction Enhancer Solution B (Toyobo, Tokyo, Japan) was used for antibody dilution. The stained retinal sections were counterstained and sealed with 4,6-diamidine 2-phenylindoledihydrochloride (DAPI)-Fluoromount-G Mounting Medium (Southern Biotech, Birmingham, AL, USA).

A confocal laser microscope (LSM 710; Carl Zeiss) was used to collect images of TUNEL staining and immunohistochemical staining.

### 4.8. Statistical Analyses

All data were presented as mean ± standard error of the mean (SEM). Student’s *t*-test was used to compare the means of two groups. One-way ANOVA followed by a Tukey–Kramer test and Dunnett’s test were used for multiple comparisons. Differences were considered to be statistically significant if *p* < 0.05.

## Figures and Tables

**Figure 1 ijms-25-01106-f001:**
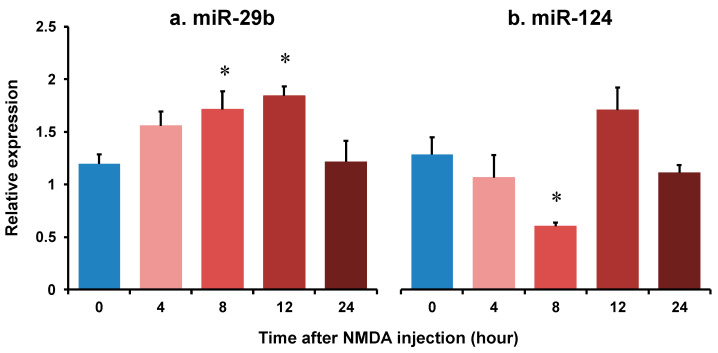
The changes in the relative expression of miR-29b (**a**) and miR-124 (**b**) in the retina of after intravitreal NMDA injection. Each datum is expressed as mean ± SEM of 4 (8 h) or 5 (other time points) independent experiments. * *p* < 0.05 vs. 0 h (before NMDA injection) using Dunnett’s test.

**Figure 2 ijms-25-01106-f002:**
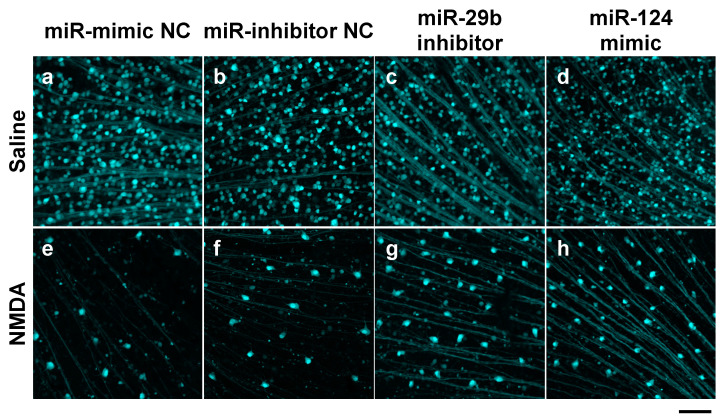
RGCs expressing ECFP in the retina of the B6.Cg-TgN(Thy1-CFP)23Jrs/J transgenic mouse. The images of whole mount of the retina of the eyes treated with 10 nmol/eye miR mimic NC and saline (**a**), 10 nmol/eye miR inhibitor NC and saline (**b**), 10 nmol/eye miR-29b inhibitor and saline (**c**), 10 nmol/eye miR-124 mimic and saline (**d**), 10 nmol/eye miR mimic NC and 40 nmol/eye NMDA (**e**), 10 nmol/eye miR inhibitor NC and 40 nmol/eye NMDA (**f**), 10 nmol/eye miR-29b inhibitor and 40 nmol/eye NMDA (**g**), and 10 nmol/eye miR-124 mimic and 40 nmol/eye NMDA (**h**) were taken with a confocal laser microscope. miR inhibitor/mimic and their NC were injected intravitreally 18 h before intravitreal NMDA injection. The retinas were excised 7 days after NMDA injection. Scale bar is 100 μm. Original magnification, ×200.

**Figure 3 ijms-25-01106-f003:**
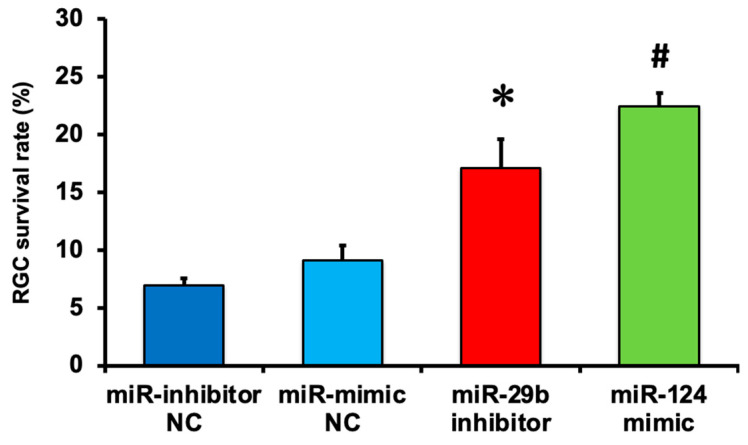
The survival rate of RGCs expressing ECFP in the retina of the B6.Cg-TgN(Thy1-CFP)23Jrs/J transgenic mouse 7 days after NMDA injection. The number of RGCs in the retina of the NMDA-injected eye was divided by that of the contralateral eye, and a percentage of RGC survival rate was calculated. Each datum is expressed as mean ± SEM of 4 (miR mimic NC), 5 (miR inhibitor NC), or 6 (miR-29b inhibitor and miR-124 mimic) independent experiments. * *p* < 0.05 vs. miR inhibitor NC; ^#^
*p* < 0.05 vs. miR mimic NC using Tukey–Kramer test.

**Figure 4 ijms-25-01106-f004:**
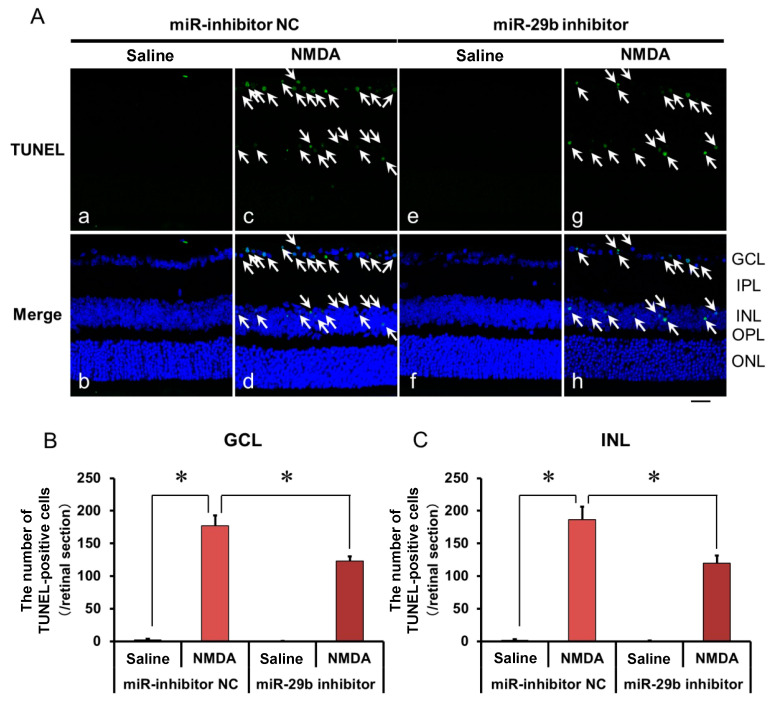
(**A**) Photomicrographs showing TUNEL staining of the retinas. The eyes were excised 12 h after intravitreal injection of saline (vehicle) or 40 nmol/eye NMDA. The eyes were treated with 10 nmol/eye miR inhibitor NC and saline (**a**,**b**), 10 nmol/eye miR-29b inhibitor and saline (**e**,**f**), 10 nmol/eye miR inhibitor NC and 40 nmol/eye NMDA (**c**,**d**), and 10 nmol/eye miR-29b inhibitor and 40 nmol/eye NMDA (**g**,**h**). miR inhibitor and its NC were injected intravitreally 18 h before intravitreal NMDA injection. Panels (**b**,**d**,**f**,**h**) (Merge) show the TUNEL staining merged with nuclear counterstaining with DAPI. GCL: the ganglion cell layer, IPL: the inner plexiform layer, INL: the inner nuclear layer, OPL: the outer plexiform layer, and ONL: the outer nuclear layer. The nuclei of TUNEL-positive cells are stained green. All the nuclei are counterstained blue. The white arrows represent the TUNEL-positive cells. Scale bar is 50 μm. Original magnification, ×400. (**B**,**C**) Effect of miR-29b inhibitor on the increase in the number of TUNEL-positive cells induced by NMDA, examined 12 h after intravitreal NMDA injection. The numbers of the TUNEL-positive cells in GCL (**B**) and INL (**C**) per one retinal section are shown. Each datum is presented as mean ± SEM of 5 independent experiments. * *p* < 0.05 between the indicated pairs using Tukey–Kramer test.

**Figure 5 ijms-25-01106-f005:**
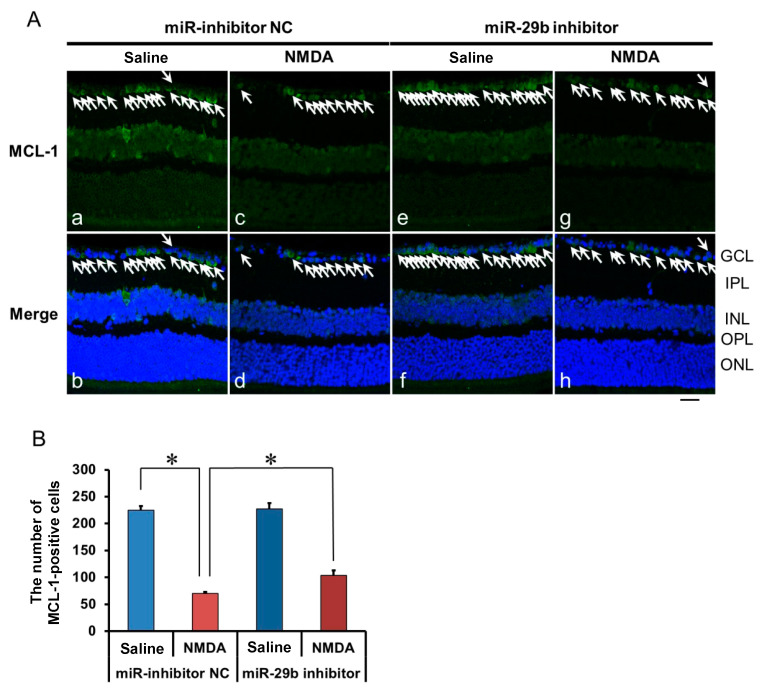
(**A**) Photomicrographs showing immunohistochemistry with anti-MCL-1 antibody of the retinas. The eyes were excised 12 h after intravitreal injection of saline (vehicle) or 40 nmol/eye NMDA. The eyes were treated with 10 nmol/eye miR inhibitor NC and saline (**a**,**b**), 10 nmol/eye miR-29b inhibitor and saline (**e**,**f**), 10 nmol/eye miR inhibitor NC and 40 nmol/eye NMDA (**c**,**d**), and 10 nmol/eye miR-29b inhibitor and 40 nmol/eye NMDA (**g**,**h**). miR inhibitor and its NC were injected intravitreally 18 h before intravitreal NMDA injection. Panels (**b**,**d**,**f**,**h**) (Merge) show the immunostaining merged with nuclear counterstaining with DAPI. GCL: the ganglion cell layer, IPL: the inner plexiform layer, INL: the inner nuclear layer, OPL: the outer plexiform layer, and ONL: the outer nuclear layer. The nuclei of MCL-1-positive cells are stained green. All the nuclei are counterstained blue. The white arrows represent the MCL-1-positive cells. Scale bar is 50 μm. Original magnification, ×400. (**B**) Effect of miR-29b inhibitor on the decrease in the number of MCL-1-positive cells induced by NMDA, examined 12 h after intravitreal NMDA injection. The number of the MCL-1-positive cells in GCL is shown. Each datum is presented as mean ± SEM of 5 independent experiments. * *p* < 0.05 between the indicated pairs using Tukey–Kramer test.

**Figure 6 ijms-25-01106-f006:**
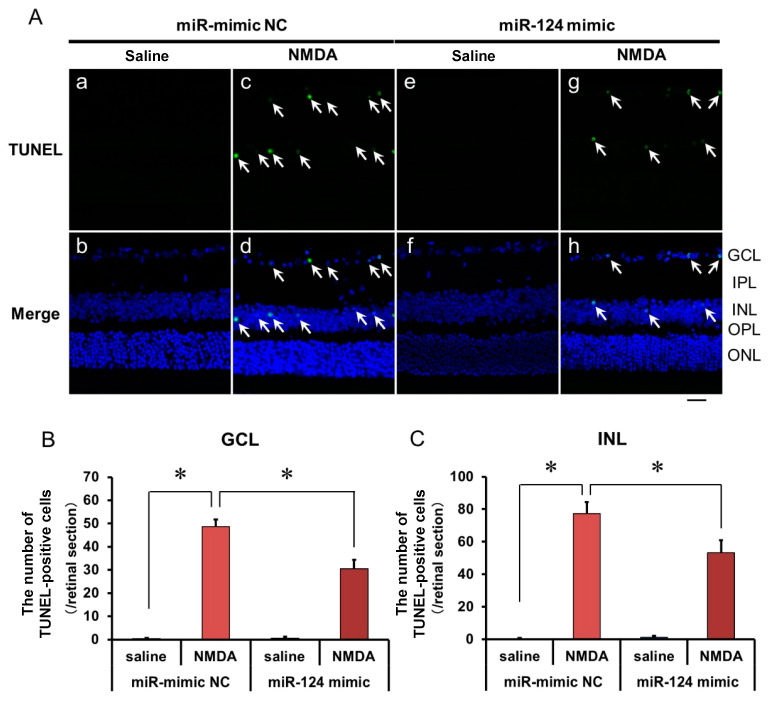
(**A**) Photomicrographs showing TUNEL staining of the retinas. The eyes were excised 12 h after intravitreal injection of saline (vehicle) or 40 nmol/eye NMDA. The eyes were treated with 10 nmol/eye miR mimic NC and saline (**a**,**b**), 10 nmol/eye miR-124 mimic and saline (**e**,**f**), 10 nmol/eye miR mimic NC and 40 nmol/eye NMDA (**c**,**d**), and 10 nmol/eye miR-124 mimic and 40 nmol/eye NMDA (**g**,**h**). miR mimic and its NC were injected intravitreally 18 h before intravitreal NMDA injection. Panels (**b**,**d**,**f**,**h**) (Merge) show the TUNEL staining merged with nuclear counterstaining with DAPI. GCL: the ganglion cell layer, IPL: the inner plexiform layer, INL: the inner nuclear layer, OPL: the outer plexiform layer, and ONL: the outer nuclear layer. The nuclei of TUNEL-positive cells are stained green. All the nuclei are counterstained blue. The white arrows represent the TUNEL-positive cells. Scale bar is 50 μm. Original magnification, ×400. (**B**,**C**) Effect of miR-124 mimic on the increase in the number of TUNEL-positive cells induced by NMDA, examined 12 h after intravitreal NMDA injection. The number of the TUNEL-positive cells in GCL (**B**) and INL (**C**) per one retinal section are shown. Each datum is presented as mean ± SEM of 5 independent experiments. * *p* < 0.05 between the indicated pairs using Tukey–Kramer test.

**Figure 7 ijms-25-01106-f007:**
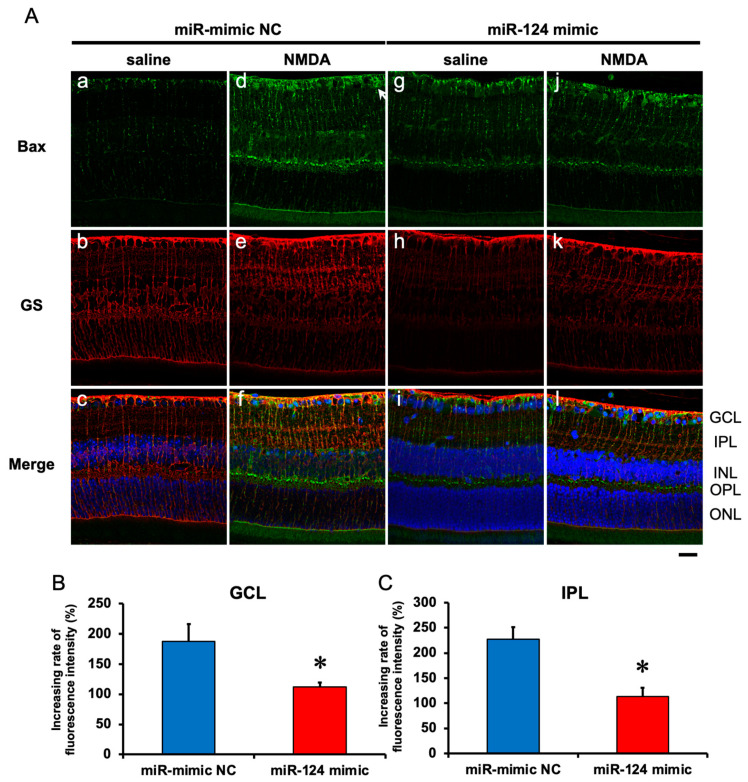
(**A**) Photomicrographs showing immunohistochemistry with anti-Bax antibody (**a**,**d**,**g**,**j**) and anti-GS antibody (**b**,**e**,**h**,**k**) of the retinas. The eyes were excised 12 h after intravitreal injection of saline (vehicle) or 40 nmol/eye NMDA. The eyes were treated with 10 nmol/eye miR mimic NC and saline (**a**,**b**,**c**), 10 nmol/eye miR-124 mimic and saline (**g**,**h**,**i**), 10 nmol/eye miR mimic NC and 40 nmol/eye NMDA (**d**,**e**,**f**), and 10 nmol/eye miR-124 mimic and 40 nmol/eye NMDA (**j**,**k**,**l**). miR mimic and its NC were injected intravitreally 18 h before intravitreal NMDA injection. Panels (**c**,**f**,**i**,**l**) (Merge) show the immunostainings with anti-Bax antibody (green) and anti-GS antibody (red) merged with nuclear counterstaining with DAPI (blue). GCL: the ganglion cell layer, IPL: the inner plexiform layer, INL: the inner nuclear layer, OPL: the outer plexiform layer, and ONL: the outer nuclear layer. Scale bar is 50 μm. Original magnification, ×400. (**B**,**C**) Effect of miR-124 mimic on the increase in the expression of Bax induced by NMDA, examined 8 h after intravitreal NMDA injection. The increasing rates of fluorescent intensity by NMDA in GCL (**B**) and IPL (**C**) are shown. Each datum is presented as mean ± SEM of 5 independent experiments. * *p* < 0.05 vs. miR mimic NC.

**Figure 8 ijms-25-01106-f008:**
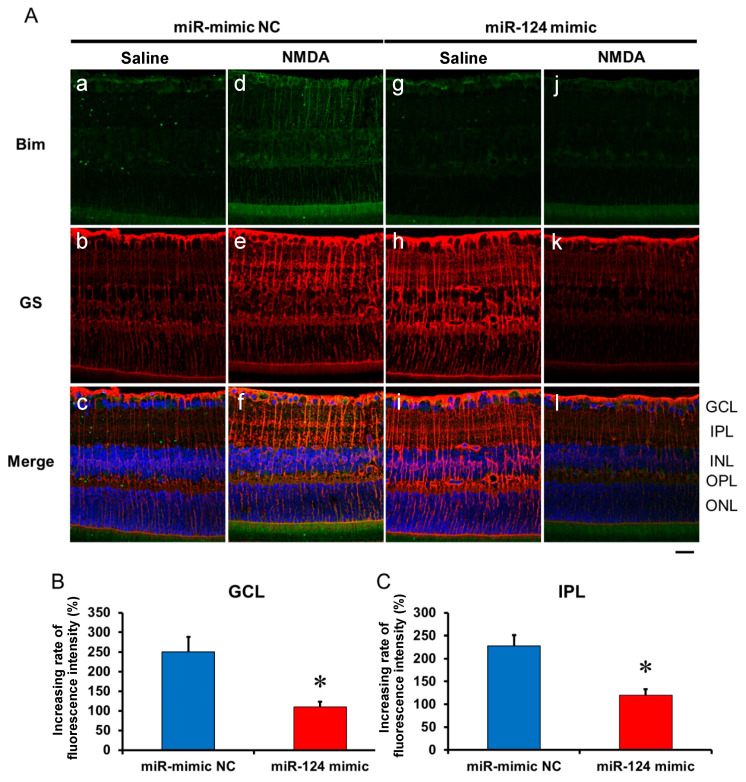
(**A**) Photomicrographs showing immunohistochemistry with anti-Bim antibody (**a**,**d**,**g**,**j**) and anti-GS antibody (**b**,**e**,**h**,**k**) of the retinas. The eyes were excised 12 h after intravitreal injection of saline (vehicle) or 40 nmol/eye NMDA. The eyes were treated with 10 nmol/eye miR mimic NC and saline (**a**,**b**,**c**), 10 nmol/eye miR-124 mimic and saline (**g**,**h**,**i**), 10 nmol/eye miR mimic NC and 40 nmol/eye NMDA (**d**,**e**,**f**), and 10 nmol/eye miR-124 mimic and 40 nmol/eye NMDA (**j**,**k**,**l**). miR mimic and its NC were injected intravitreally 18 h before intravitreal NMDA injection. Panels (**c**,**f**,**i**,**l**) (Merge) show the immunostainings with anti-Bim antibody (green) and anti-GS antibody (red) merged with nuclear counterstaining with DAPI (blue). GCL: the ganglion cell layer, IPL: the inner plexiform layer, INL: the inner nuclear layer, OPL: the outer plexiform layer, and ONL: the outer nuclear layer. Scale bar is 50 μm. Original magnification, ×400. (**B**,**C**) Effect of miR-124 mimic on the increase in the expression of Bim induced by NMDA, examined 8 h after intravitreal NMDA injection. The increasing rates of fluorescent intensity by NMDA in GCL (**B**) and IPL (**C**) are shown. Each datum is presented as mean ± SEM of 5 independent experiments. * *p* < 0.05 vs. miR mimic NC.

**Figure 9 ijms-25-01106-f009:**
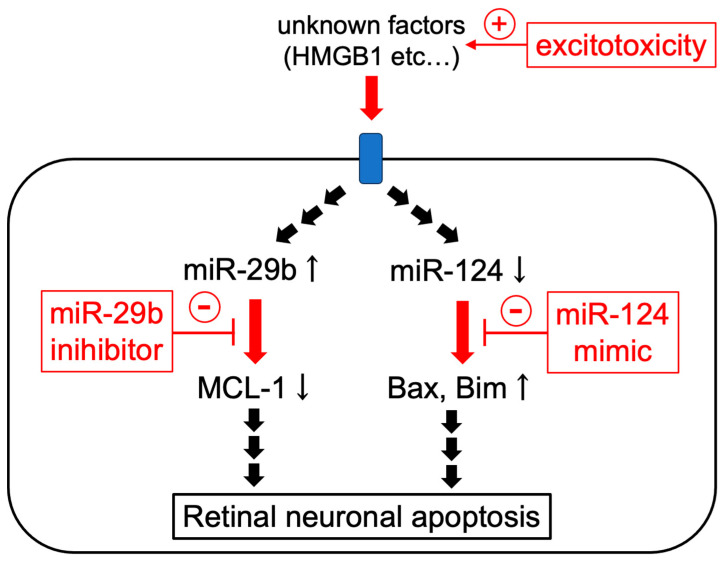
The scheme of the possible mechanisms of the protective effects of miR-29b inhibitor and miR-124 mimic.

**Table 1 ijms-25-01106-t001:** The sequences of the target-specific 5′-primers used in the present study.

Target	Sequence
Mouse miR-29b	5′-GCTGGTTTCATATGGTGGTTTA-3′
Mouse miR-124	5′-TAAGGCACGCGGTGAATGCC-3′
Mouse snoRNA234	5′-GGAACTGAATCTAAGTGATTTAACAA-3′

## Data Availability

Data are contained within the article and Appendix A.

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
