# Peer review of "The Role of microRNAs Related to Apoptosis for *N*-Methyl-d-Aspartic Acid-Induced Neuronal Cell Death in the Murine Retina"

_ijms, 2024, doi:10.3390/ijms25021106_

Round 1
Reviewer 1 Report
Comments and Suggestions for Authors
This research manuscript provides valuable insights into the role of miR-29b and miR-124 in retinal injury and apoptosis induced by intravitreal NMDA injection. However, there are several critical points and questions that should be addressed:
Comment 1: The authors mentioned that miRNA expression changes occurred 8-12 hours after NMDA injection, which aligns with the timing of apoptosis. However, can you provide insights into the underlying mechanisms or triggers for these expression changes?
Comment 2: The authors should clarify why miR inhibitor NC was chosen as a control. Were other control groups considered, such as a non-targeting miRNA control or a control without treatment?
Comment 3: Discuss the potential mechanisms through which miR-29b inhibitor reduces apoptosis of retinal neuronal cells. How do these findings relate to the broader understanding of apoptosis in retinal cells?
Comment 4: The authors should mention the number of mice used for each experiment. Additionally, it appears that there are discrepancies in Figure numbers 5, 6, and 7 regarding the intensity and the number of arrows compared to positive cell rechecks. These figures should be carefully reanalyzed and adjusted as needed for accuracy.
Comment 5: The authors should elaborate on the biological implications of the changes in MCL-1 expression induced by miR-29b inhibitor. How does MCL-1 relate to retinal injury and apoptosis?
Comment 6: The authors should provide a graphical outline diagram to provide detailed mechanistic insights into how miR-29b, miR-124, MCL-1, Bax, and Bim interact and influence retinal injury and apoptosis. This will assist readers in comprehending the molecular pathways involved.
Author Response
Thank you very much for your thoughtful comments.
Comment 1: The authors mentioned that miRNA expression changes occurred 8-12 hours after NMDA injection, which aligns with the timing of apoptosis. However, can you provide insights into the underlying mechanisms or triggers for these expression changes?
We also think that the reviewer’s question is very important. Unfortunately, we could not clarified the mechanisms and/or the triggers of the expressional changes of miRs in the present study. Recently, the high mobility group box-1 (HMGB1)-receptor of advanced glycation end-products (RAGE) axis has been involved in the regulation of miR expression (Ho et al., 2023; Ma et al., 2023). We previously reported that the HMGB1-RAGE axis was involved in neurodegeneration induced by intravitreal NMDA in the retina (Sakamoto et al., 2015). Although further studies are clearly needed, we consider the HMGB1-RAGE axis as a possible candidate of the triggers of the expressional changes of miRs. We added some description about this in Discussion.
Comment 2: The authors should clarify why miR inhibitor NC was chosen as a control. Were other control groups considered, such as a non-targeting miRNA control or a control without treatment?
Both miR mimic NC and miR inihibitor NC used in the present study are based on cel-miR-67, which is Caenorhabditis elegans miR and confirmed to have minimal sequence identity with murine miRNAs. Therefore, they are non-targeting miRNA controls. They are confirmed to have no identifiable effects on tested miRNA function by the manufacturer, Dharmacon. We added the description in Results.
Comment 3: Discuss the potential mechanisms through which miR-29b inhibitor reduces apoptosis of retinal neuronal cells. How do these findings relate to the broader understanding of apoptosis in retinal cells?
Comment 5: The authors should elaborate on the biological implications of the changes in MCL-1 expression induced by miR-29b inhibitor. How does MCL-1 relate to retinal injury and apoptosis?
This is the answer for the reviewer’s comments #3 and #5. MCL-1 is one of bcl-2 family proteins, and reduces activation of Bax/Bak, which are pro-apoptotic proteins and lead to cytochrome c release from mitochondria and then apoptosis (reviewed by Thomas et al., 2010). In the present study, we showed that miR-29b inhibitor reduced apoptosis of retinal neuronal cells (Figure 4) and upregulated MCL-1 protein expression (Figure 5). MCL-1 has been reported to be one of the targets of miR-29b. It is possible that miR-29b inhibitor reduces degradation of MCL-1 mRNA, leads to upregulation of expression of MCL-1 protein, and then represents anti-apoptotic effect. We added the description in Discussion.
Comment 4: The authors should mention the number of mice used for each experiment. Additionally, it appears that there are discrepancies in Figure numbers 5, 6, and 7 regarding the intensity and the number of arrows compared to positive cell rechecks. These figures should be carefully reanalyzed and adjusted as needed for accuracy.
We mentioned the number of mice used in Results section and Figure Legends.
We checked the figures and corrected the legend and removed the arrows in Figure 7A.
Comment 6: The authors should provide a graphical outline diagram to provide detailed mechanistic insights into how miR-29b, miR-124, MCL-1, Bax, and Bim interact and influence retinal injury and apoptosis. This will assist readers in comprehending the molecular pathways involved.
As the reviewer’s suggestion, we added the scheme of the possible mechanisms of the protective effects of miR-29b inhibitor and miR-124 mimic (Figure 8 in the revised manuscript).
Reviewer 2 Report
Comments and Suggestions for Authors
As attached

Author Response
Thank you very much for your thoughtful comments.
Introduction
Line 27: add a reference to the first sentence
We added a reference to the first sentence.
Line 53-54: “In the present study, we tried to show that excitoneurotoxicity of the retinal neurons caused expressional changes of miR-29b and miR-124.” Sound like a result. You can rephrase as “The work is aimed at showing the correlation between excitoneurotoxicity of the retinal neurons and expressional changes of miR-29b and miR-124”.
We changed the text as the reviewer suggested.
Results
Line 63: “1.84 ± 0.08 and 1.84 ± 0.08” is confusing. Were the increments the same at 8 and 12 hours respectively? The graph does not suggest so. Amend this accordingly
We are sorry about the mistake. We corrected the value at 8 hours after NMDA injection.
The expression level of miR-29b was increased to 1.71 ± 0.18 and 1.84 ± 0.08, 8 and 12 hours after NMDA injection respectively (Figure 1a).
Line 68: For the tended increase, insert the p value
We inserted the p value.
Line 150: instead of always using clarified, use “demonstrated”. It is more scientific. Apply same to other places
We changed the text as the reviewer suggested.
Discussion
Line 262: The data should be shown as supplementary data
We showed the data in Supplemental Figure 1.
Line 267: Who are the other researchers? The authors should be cited appropriately
We inserted the citation numbers.
Line 314: Show the data in the supplementary section
We showed the data in Supplemental Figure 2.
Line 358: Cite the author of the method and perhaps state the formular
We cited the reference.
Line 426: All data were presented…
We changed the text as the reviewer suggested.
Materials and methods
Line 427: An ANOVA should be done to determine the overall variation among all the groups before the use of Tukey–Kramer test to find where the significance lies between groups
We did ANOVA followed by multiple comparisons. We corrected the text as the reviewer’s comment.
General comments
Inadequate references in the discussion. In discussing a work, it is important to compare your work with other works. The more, the merrier.
We added some discussion about the relationship between miR and the retinal diseases. As far as we know, there is only one paper that was shown the relationship between miR and excitotoxicity in the retina.
Reviewer 3 Report
Comments and Suggestions for Authors
In the manuscript presented, the authors demonstrated that upregulation of miR-29b and downregulation of miR-124 in the retina of the NMDA-injected eyes in the mice, in vivo. Intravitreal injection of miR-29b inhibitor and miR-124 mimic reduced RGC death. They reduced downregulation of MCL-1, an anti-apoptotic factor, and upregulation of Bax and 255 Bim, pro-apoptotic factors, in the retina induced by intravitreal NMDA. These results suggest that expressional changes of microRNA are involved in excitotoxicity of RGC, and that miR-29b inhibitor and miR-124 prevented NMDA-induced retinal injury via reduction of downregulation of pro-apoptotic proteins and upregulation of pro-apoptotic proteins.
The paper is well written and logically substantiated. The history of the problem is described; discussion is also relevant to the findings, though should be improved by recent papers published for the last 5 years (just one papers for this period from 2019 is considered by authors). My minor notes are related to the absence of deciphering of some abbreviations (e.g. NC, though it is clear for professionals to be negative control), and the list of abbreviations should be inserted at the end of the manuscript. Also, the origin of reagents needs to be pointed out (NMDA etc).
Author Response
Thank you very much for your thoughtful comments.
We added recent literatures reported about the relationship between miR and retinal function and/or pathology in Discussion.
We added the full spell of every abbreviation in the text and the list of abbreviations was inserted at the end of the manuscript (Table 2 in the revised manuscript).
In addition, the origins of the reagents used in the present study were added in the text.
Round 2
Reviewer 1 Report
Comments and Suggestions for Authors
The author has incorporated comments, replied, and provided justifications well within the manuscript.